# Molecular Characterization of Multidrug-Resistant and Hypervirulent New Delhi Metallo-Beta-Lactamase *Klebsiella pneumoniae* in Lazio, Italy: A Five-Year Retrospective Study

**DOI:** 10.3390/antibiotics13111045

**Published:** 2024-11-05

**Authors:** Claudia Rotondo, Carolina Venditti, Ornella Butera, Valentina Dimartino, Francesco Messina, Michele Properzi, Claudia Caparrelli, Valentina Antonelli, Silvia D’Arezzo, Marina Selleri, Carla Nisii, Carla Fontana

**Affiliations:** National Institute for Infectious Diseases “L. Spallanzani” Istituto di Ricovero e Cura a Carattere Scientifico, via Portuense 292, 00149 Rome, Italy; claudia.rotondo@inmi.it (C.R.); carolina.venditti@inmi.it (C.V.); ornella.butera@inmi.it (O.B.); valentina.dimartino@inmi.it (V.D.); francesco.messina@inmi.it (F.M.); michele.properzi@inmi.it (M.P.); claudia.caparrelli@gmail.com (C.C.); valentina.antonelli@inmi.it (V.A.); silvia.darezzo@inmi.it (S.D.); marina.selleri@inmi.it (M.S.); carla.fontana@inmi.it (C.F.)

**Keywords:** carbapenem-resistant, *klebsiella pneumoniae*, molecular typing, NDM, virulence genes, whole-genome sequencing

## Abstract

Background/Objectives: Antimicrobial resistance represents a challenge to public health systems because of the array of resistance and virulence mechanisms that lead to treatment failure and increased mortality rates. Although for years the main driver of carbapenem resistance in Italy has been the *Klebsiella pneumoniae* KPC carbapenemase, recent years have seen an increase in VIM and NDM metallo-beta-lactamases (MBLs). We conducted a five-year survey of New Delhi Metallo-beta-Lactamase (NDM)-producing *Klebsiella pneumoniae* (NDM-Kpn) clinical isolates from the Lazio region, Italy; the study aimed to elucidate the molecular mechanisms underpinning their resistant and virulent phenotype. Methods: Antimicrobial susceptibility was evaluated by automated systems and broth microdilution. In silico analysis of acquired resistance and virulence genes was performed using whole-genome sequencing (WGS), molecular typing through MLST, and core genome multi-locus sequence typing (cgMLST). Conclusions: A total of 126 clinical NDM-Kpn isolates were collected from 19 distinct hospitals in the Lazio region. Molecular analysis highlighted the existence of NDM-1 (108/126) and NDM-5 (18/126) variants, 18 Sequence Types (STs), and 15 Cluster Types (CTs). Notably, 31/126 isolates displayed a virulence score of 4, carrying *ybt*, *ICEKp*, *iuc*, and *rmp* genes. This study identified a variety of NDM-Kpn STs, mainly carrying the *bla*_NDM-1_ gene, with a significant number linked to high-risk clones. Of these isolates, 24.6% showed high-level resistance and virulence, emphasizing the risk of the spread of strains that combine multi-drug-resistance (MDR) and virulence. Proactive surveillance and international collaborations are needed to prevent the spread of high-risk clones, as well as further research into new antimicrobial agents to fight antibiotic resistance.

## 1. Introduction

The global rise in antimicrobial resistance represents a formidable challenge to public health systems worldwide. Among the most concerning pathogens are carbapenem-resistant *Enterobacteriaceae* (CRE), particularly carbapenem-resistant *Klebsiella pneumoniae* (CR-*K. pneumoniae*), which are the primary culprits behind severe and often untreatable hospital-acquired bacterial infections [1]. The emergence of these pathogens is characterized by an alarming array of resistance and virulence mechanisms, leading to treatment failure and increased mortality rates [2].

In Italy, for many years, the primary drivers of carbapenem resistance have been the serine-beta-lactamases, especially the *Klebsiella pneumoniae* carbapenemases (KPCs). Nonetheless, shifts in the epidemiology of nosocomial infections and colonization by CRE have been documented, indicating an escalation in metallo-beta-lactamases (MBLs) VIM and NDM, along with the propagation of other designated high-risk clones (e.g., Sequence Types (STs) 512, 101, 147, and 307) [3,4]. NDM enzymes, in particular, are the most prevalent type of MBL among carbapenemase-producing *Enterobacteriaceae*, and have a remarkable capacity for horizontal dissemination, making them difficult to control [5]. Since their initial description, several variants of NDM-type enzymes have been identified in different strains from across all continents, highlighting the global challenge posed by these resistant bacteria [6,7,8]. 

Italy has also been affected by the spread of NDM-type enzymes. The first detection of the NDM-1 variant in Italy occurred in 2009 in two patients hospitalized in Modena. Epidemiological studies linked these cases to a third patient with a history of hospital admission in India [6]. 

One of the most severe outbreaks recorded in the country, involving almost 1645 cases, occurred in Tuscany and was mainly associated with NDM-1 producing *K. pneumoniae* with the most dominant ST147 [9]. Furthermore, in 2020, ST147 NDM-9 producing *K. pneumoniae* was found in two critically ill patients [10]. 

Due to their high circulation, the European Centre for Disease Prevention and Control (ECDC) issued a document in June 2019 calling for actions to prevent the spread of NDM-producing *K. pneumoniae* (NDM-Kpn) isolates [11]. In the framework of a regional surveillance scheme aiming at collecting all ceftazidime-avibactam (CZA)-resistant isolates, including naturally resistant NDM-Kpn isolates, in our capacity as a regional reference center [12], we were able to analyze a considerable number of NDM-Kpn isolates coming from various hospitals in our region. Additionally, based on the recent literature showing the increasing impact of the convergence of resistance and virulence in *K. pneumoniae* strains leading to the formation of hypervirulent (hvKp) and multidrug-resistant (MDR) strains [13,14,15], we aimed to investigate the presence of MDR hvKp profiles. The potential of specific hvKp strains in posing a significant healthcare risk when coupled with carbapenem resistance was also highlighted in a recent alert from the ECDC [16]. Although hvKp isolates are rarely resistant, being associated with liver abscesses in otherwise healthy individuals [17,18], more recent reports showed changes in susceptibility patterns of hvKp, when strains carrying ESBLs or carbapenemases began to emerge [19,20,21]. 

Given the above findings and our role as regional reference center, the objective of our study was to delineate the epidemiological, genotypic, and virulence characteristics of NDM-Kpn strains collected between January 2019 and December 2023 in various hospitals of the Lazio Region.

## 2. Results

### 2.1. Phenotypic Characteristics

A total of 1064 strains were collected for this study. All isolates were resistant to carbapenems, piperacillin/tazobactam, and ciprofloxacin, while 79.4% of isolates (100/126) were resistant to gentamicin, 74.6% (94/126) to trimethoprim/sulfamethoxazole, and 14.3% (18/126) to colistin (Appendix A).

Of the isolates studied, 448 were shown to produce a serine carbapenemase (comprising 439 KPC and 9 OXA-48-like), 12 resulted in ESBL CTX-M-15-producers, and 604 isolates were found to carry a MBL (326 VIM and 278 NDM). Among the NDM-carrying *Enterobacterales*, 168 were *K. pneumoniae*. Out of the total, 126 strains (one isolate per patient) were analyzed by WGS. The isolates were obtained from different specimens, including 48% from rectal swabs (60/126), 25% from urine (32/126), 10.4% from blood (13/126), 7% from wound swabs (9/126), 4% from bronchoalveolar lavage (5/126), 1.6% each from bronchial aspirate and drainage (2/126 each), and finally 0.8% each from urethral swabs, vaginal swabs, and femoral catheters (1/126 each) (Appendix A).

### 2.2. Genomic Characterization of Resistance and Virulence Profile in NDM-Kpn Isolates

Analysis of the resistome revealed the presence of the *bla*_NDM-1_ (108/126) and *bla*_NDM-5_ (18/126) gene variants. As a single resistance determinant, the NDM carbapenemase was detected in 16/126 isolates, in association with a serine beta-lactamase OXA-48-like (22/126), KPC (5/126), SCO-1 (1/126), and with the plasmid-encoded class C beta-lactamases CMY-2, CMY-6, and DHA-1 in 1, 11, and 3 isolates, respectively. The co-production of an ESBL *bla*_CTX-M-15_, *bla*_CTX-M-14b_, and *bla*_LAP-2_ was identified in 98, 4, and 1 isolates, respectively. The MBL VIM-2 was detected in seven isolates. Additional resistance genes, conferring resistance to sulphonamides, trimethoprim, aminoglycosides, quinolones, and other classes of antibiotics, are shown in Appendix A.

Genotyping analysis showed that the 126 NDM-Kpn strains were associated with 18 different STs. The most common ST was ST147, accounting for 44% of the isolates (55/126), followed by ST11 at 23% (29/126) and ST395 at 7% (9/126). Other STs were present in smaller proportions as follows: 5.5% for ST1805 (7/126), 4.8% for ST383 (6/126), 4% for ST15 (5/126), and 3% for ST512 (4/126). All of the remaining STs, such as ST14, 17, 23, 29, 234, 307, 1117, 3299, 4081, 4053, and 6118 comprised only one isolate (Table 1).

Based on the cgMLST results, the isolates were categorized into 15 different Cluster Types (CTs) (CT-1 to CT-15) (Figure 1 and Table 1). Specifically, ST147 clustered into five different CTs (CT-1, CT-2, CT-3, CT-4, and CT-5), while ST11, ST15, ST383, and ST395 were grouped into two (CT-9 and CT-10, CT-14 and CT-15, CT-12 and CT-13, and CT-7 and CT-8, respectively). Isolates from ST512 and ST1805 formed a single cluster, CT-11 and CT-6, respectively.

As shown in Table 1, the NDM-1 strains belong to fourteen different STs and the NDM-5 strains to seven. Both variants are widely spread among different hospitals in the Lazio region. Among the NDM-1 strains, 42.6% (46/108) are associated with ST147. These isolates are divided into CT-1 (13/46), CT-3 (27/46), CT-4 (4/46), and CT-5 (2/46). Additionally, 24.1% of isolates (26/108) are linked to ST11, with 25 of these falling into CT-10. Moreover, 8.4% of ST395 strains (9/108) are grouped into CT-8 (4/9) and CT-7 (3/9), while the other two do not form a CT. ST1805 accounts for 6.5% of the strains (7/108), with six falling into CT-6 and one not forming a CT. Notably, four strains of CT-4 and three ST1805 strains are associated with the same hospital (H-3 and H-1, respectively). ST15 comprises 4.7% of the isolates (5/108), with three grouped in CT-14, all originating from hospital H-3, and the remaining two clustered in CT-15. STs 512 and 383 each comprise 3.7% of the strains (4/108 each). The former (ST512) forms a single cluster (CT-11), with three of the four strains originating from hospital H-9, while the latter (ST383) falls into two CTs (CT-13 with three strains and CT-12 with one). Lastly, the remaining seven strains are each associated with a different ST (ST17, ST23, ST29, ST307, ST1117, ST3299, and ST4053) and do not form a CT. Regarding the NDM-5 variant, again 50% of the strains (9/18) belong to ST147, of which five cluster in CT-2, two in CT-3 and two do not form CTs. Both the CT-2 and CT-3 strains were derived from the same hospital, H-9 and H-7, respectively. In addition, 16.7% of strains (3/18) belong to ST11. Interestingly, they are all grouped in CT-9 and were derived from the same hospital (H-2). ST383 comprises 11.1% of isolates (2/18), of which one clustered in CT-2, and the other did not form a CT. The remaining four strains belong to ST14, ST234, ST4081, and ST6118, without forming CTs.

It is interesting to note that the two isolates that form CT-12 (NDM Kpn-5 and NDM Kpn-11) carry different NDM variants, despite coming from the same hospital (H-3) (one that receives patients from North Africa); in one isolate, the NDM-5 variant was found, associated with CTX-M-15 and minor oxacillinases, while in the other, the NDM-1 variant was combined with OXA-48, CMY-6, and CTX-M-15.

Considering all isolates and their virulence profiles, it is noteworthy that 24.6% (31/126 isolates) exhibited a virulence score of 4 (Table 1). All isolates carried aerobactin and yersiniabactin genes (*iuc* and *ybt*), as well as hypermucoviscosity regulatory genes (*rmpA* and *rmpA2*). Specific STs associated with virulence were ST147 KL64 (*wzi*64), ST395 KL2 (*wzi*2), and ST15 KL112 (*wzi*93).

Data on CTs, virulence score, and hospitals where the isolates were grown are summarized in Table 2. It is worth noting that the highest virulence score was found in CT-3, where 19/25 isolates (all belonging to ST147) had a virulence score of 4, and were obtained from 13 different hospitals in the wider Rome area; this suggests the widespread distribution of this clone. In contrast, CT-10 was characterized by 25 isolates that displayed a lower virulence score.

Table 2 also shows that a few CTs (CT-14 and CT-7) comprised three isolates, each with the highest virulence score of 4, derived from the same hospitals (H-3 and H-1). Among the remaining ninety-five isolates, nine had a score of 3, while the rest scored between 0 and 1, which was considered insignificant.

## 3. Discussion

CR-*K. pneumoniae* is a significant pathogen responsible for causing various human infections. The global spread and increasing incidence of CR-*K. pneumoniae* strains have posed significant challenges to treatment, presenting a major threat to human health and contributing to increased mortality rates [22]. Of particular concern is MBL-producing *K. pneumoniae*, as there are limited therapeutic options available for treating infections caused by this strain. In fact, despite the introduction of several beta-lactamase inhibitors, such as CZA, these combinations are ineffective against strains producing MBLs [5].

The epidemiology of MBL-producing bacteria is evolving, with a progressive spread of NDM-carrying bacteria beyond the Indian subcontinent. Today, at least 70 NDM variants are known (https://www.ncbi.nlm.nih.gov, accessed on 26 September 2024) and have been reported across all continents, highlighting the global challenge posed by these resistant bacteria [6]. In an effort to combat the spread of high-threat pathogens, an active surveillance for the emergence of resistance to CZA in *Enterobacteriaceae* (including intrinsically resistant strains such as MBLs) has been set up (beginning in 2019) at the “Lazzaro Spallanzani” National Institute for Infectious Diseases I.R.C.C.S. (INMI), which serves as a regional reference center for other hospitals in the area.

Our results show the presence of combined resistance in 111/126 NDM isolates with other ESBLs and carbapenemases; this is not surprising and reflects the complex challenge of MDR bacteria [5,23,24]. The most frequent combination was with CTX-M-15, likely linked to the widespread diffusion of CTX-M-15 associated with numerous high-risk clones [25]. Additionally, 17.5% of isolates (22/126) displayed an association of NDM with class D carbapenemases OXA-48, which have become endemic in North Africa, the Middle East, and China, with a continuous increase reported in European countries [26]. KPC-3, endemic in Italy since 2013 [27], was found in only five isolates in our study, three of which came from the same hospital (H-9), indicating a more localized spread.

The resistome analysis of the 126 NDM-Kpn strains revealed that, as expected, the NDM-1 and NDM-5 variants were the most prevalent [28]. NDM-1 has often been identified in Italy, even in isolates responsible for outbreaks [5,29,30,31]. The NDM-5 variant, identified in only 15% of isolates, is an allelic variant most commonly found in China and areas of Africa, indicating the rapid spread and circulation of these strains [32].

Phylogenetic analysis (Figure 1 and Appendix A) identified 18 different STs demonstrating the genetic variability and widespread nature of the *bla*_NDM_ gene, which is located on different transposons and plasmids and is not strictly related to a single ST [33].

The presence of virulence genes in the isolates with an MDR profile is another growing concern [34]. In our study, approximately 25% of the isolates exhibited a high virulence score of 4 and were closely associated with three STs widely spread among the hospitals involved in the study (Table 1 and Table 2). Among the virulence determinants detected are the yersiniabactin (*ybt*) and aerobactin (*iuc*) genes, which are primarily responsible for the high virulence of hvKp isolates, conferring a high potential to spread in healthcare settings and higher morbidity and mortality [35]. No evident correlation was found between the presence of resistant genes vs. phenotypic resistance (Appendix A), indicating a disconnect between genotype and phenotype, which has already been described [36].

Since 2009, Italy has recorded cases involving the NDM-1 variant in Modena, where these cases were connected to a third patient returning from India, indicating that international travel may have played a role in the spread of this resistance [6]. In 2011, the NDM-1 variant was detected in *K. pneumoniae* and *E. coli* isolates from six patients hospitalized in the Bologna area, Northern Italy. This discovery began a series of outbreaks in the region and beyond, culminating in a large outbreak of NDM-CRE in Tuscany in 2018. The outbreak was associated with the clonal expansion of the NDM-1-producing ST147 lineage of *K. pneumoniae*, with minor cases involving *E. coli* and other lineages of NDM-Kpn type enzymes [5].

Our results are in line with these previous reports from other Italian regions, as the local epidemiology of NDM-Kpn highlighted by our study reflects the prevalence of the main clones involved in recent Italian outbreaks, such as ST147, ST11, ST395, and ST15 [5,37]; however, the regional nature of our study (which involved the Lazio region only) remains an important limitation of our work. Another weakness is the possible underestimation of the number of NDM-Kpn isolated, since not all regional laboratories have the means to characterize their isolates in terms of their mechanism of resistance, and therefore some NDM strains may have been missed. We also observed a high variability in the number of strains received from other laboratories between 2019 and 2023 (Appendix A), with much higher numbers received in 2022 and 2023, which does not allow a reliable analysis of changing trends during the study period. Being a laboratory surveillance study, our research also lacks clinical data of patients and treatment outcomes, which would have allowed us to gain insight into the real impact of virulent strains and their therapeutic challenges.

It is clear from our results, however, that the increasing occurrence of these pathogens underscores the need for rapid and effective strategies to identify and monitor their spread. In our study, the use of advanced molecular technologies, such as WGS, allowed us to highlight not only the presence of these MDR strains but also the real impact of these strains (in terms of STs and clusters detected), their level of risk (deriving from the virulence of the strains involved), and their presence in different hospitals.

In conclusion, our findings shed some light on the resistance and virulence profiles of NDM-Kpn circulating in our geographical area. Further studies are needed to deepen our understanding of the resistome and virulome of MDR microorganisms, their clinical significance, and the therapeutic challenges they pose.

Since the identification of virulent and hypervirulent phenotypes represents an unmet diagnostic need, the development of rapid diagnostic systems, such as real-time PCRs for their early detection, would be highly beneficial for effective control strategies.

## 4. Materials and Methods

### 4.1. Collection of Clinical Isolates

From January 2019 to December 2023, we implemented a comprehensive surveillance program to collect Gram-negative *Enterobacterales* clinical strains from patients that showed resistance to CZA, including naturally resistant isolates producing MBLs such as NDM and VIM. These isolates were grown from patients treated at different hospitals in the city and the Lazio region. We conducted further evaluations in our laboratory to assess their antibiotic susceptibility and to perform phenotypic and molecular characterization. Only NDM-Kpn isolates confirmed to produce the NDM carbapenemase underwent additional molecular characterization.

### 4.2. Phenotypic and Molecular Characterization of Isolates

Antimicrobial susceptibility testing and species identification were performed using the Phoenix system (Becton Dickinson Diagnostics, San Jose, CA, USA) and the MALDI-TOF Biotyper sirius System (Bruker Daltonics, Bremen, Germany), respectively. The determination of minimum inhibitory concentrations (MICs) for colistin was executed through broth microdilution (Liofilchem, Roseto degli Abruzzi, Italy). All results were interpreted following the recent guidelines established by the European Committee on Antimicrobial Susceptibility Testing (EUCAST) [38] and the MIC breakpoint used for CZA was 8 mg/L. As resistance to CZA could be due to the presence of a mutated KPC or a metallo-beta-lactamase, our initial identification of MBL NDM involved the use of a lateral flow immunochromatography assay (NG-Test CARBA 5, Biotech, Paris, France), This was subsequently validated through whole-genome sequencing (WGS) using Illumina Miseq (Illumina, San Diego, CA, USA). In silico analysis of the sequence data were conducted employing dedicated tools. The identification of STs and resistance profiles was facilitated by the ResFinder v3.0 web server (http://www.genomicepidemiology.org, accessed on 1 September 2024). The threshold for a minimum percentage of sequence identity was established at 100%, with a required alignment length of >98%.

The virulence of the isolates was assessed using Kleborate v 2.0.4 (https://github.com/klebgenomics/Kleborate, accessed on 1 September 2024). This software is specifically designed for *K. pneumoniae* and examines five main acquired virulence loci that are commonly found among hvKp strains. These loci include the siderophores yersiniabactin (*ybt*), aerobactin (*iuc*), and salmochelin (*iro*), the genotoxin colibactin (*clb*), and the hypermucoid locus *rmpADC*. Using the software, we calculated virulence scores ranging from 0 to 5 based on the presence of virulence genes. Additionally, the Kleborate software predicted the K and O antigen serotypes and the *wzi* allele. To further explore genetic relationships, we used the WGS-based core genome MLST (cgMLST) scheme v1.0, employing the Ridom SeqSphere+ software (Ridom GmbH, Münster, Germany) with default settings. Using the defined *K. pneumoniae sensu lato* cgMLST (comprising 2358 target genes), we compared genomes using a gene-by-gene approach [39]. Compared to the reference strain (GenBank accession no. NC_012731), the resulting set of target genes was used to interpret the clonal relationship displayed in a minimum spanning tree. Genotypically related isolates (with a distance of ≤15 alleles) were identified within a CT (https://www.cgmlst.org/ncs, accessed on 1 September 2024). All raw reads generated were submitted to the Sequence Read Archive (SRA) under BioProjects ID PRJNA686854 and ID PRJNA1125835.

## 5. Conclusions

A critical point has been reached in the ongoing battle against CR-*K. pneumoniae*, and the spread of NDM-producing strains with a hypervirulent profile underscores the necessity of a global response following the One Health approach, which integrates surveillance at the hospital as well as community and environmental settings, further research, policy making, and the implementation of national and international guidelines.

The story of NDM-carrying bacteria and their spread through various continents reasserts the need for international collaborations to address the threat of emerging infectious agents. Moving forward, healthcare systems worldwide should receive adequate resources to adopt more robust infection control measures, and more funding should be directed towards developing new antibiotics and treatment options to win the fight against these ever-evolving pathogens.

## Figures and Tables

**Figure 1 antibiotics-13-01045-f001:**
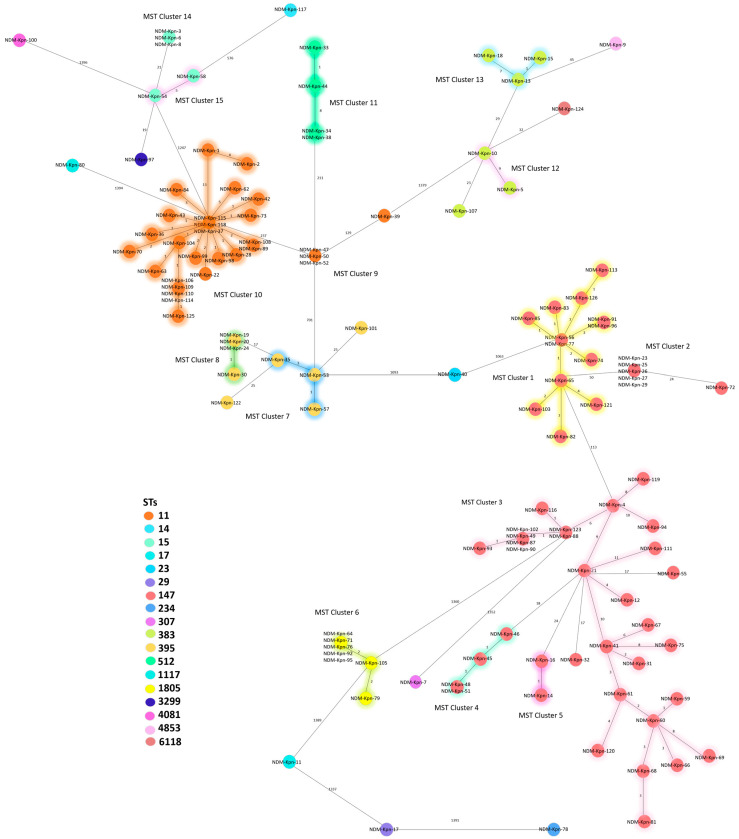
Minimum Spanning Tree (MST) of 126 NDM-Kpn isolates. The numbers indicate the allelic distance between different strains or clusters, based on cgMLST analysis with RIDOM SEQSPHERE software, version 9.0.10 2023-09. Cluster distance threshold: 15 alleles.

**Table 1 antibiotics-13-01045-t001:** Genotyping analysis, molecular typing and characterization of virulence genes of NDM-Kpn.

Strain	Hospital ^1^	NDM Variant	Typing				Virulence Determinants	
			ST ^2^	CT ^3^	*wzi* Allele	KL ^4^	Virulence Genes	Score ^5^
NDM-Kpn-1	H-4	*bla* _NDM-1_	11	10	*wzi*24	KL24	*ybt15*, *ICEKp11*	1
NDM-Kpn-2	H-1	*bla* _NDM-1_	11	10	*wzi*24	KL24	*ybt15*, *ICEKp11*	1
NDM-Kpn-3	H-3	*bla* _NDM-1_	15	14	*wzi*93	KL112	*ybt16*, *ICEKp12*, *iuc1*, *rmp1*, *KpVP-1*	4
NDM-Kpn-4	H-3	*bla* _NDM-1_	147	3	*wzi*64	KL64	*ybt9*, *ICEKp3*, *iuc1*, *rmp*	4
NDM-Kpn-5	H-3	*bla* _NDM-5_	383	12	-	KL30	-	3
NDM-Kpn-6	H-3	*bla* _NDM-1_	15	14	*wzi*93	KL112	*ybt16*, *ICEKp12*, *iuc1*, *rmp*	4
NDM-Kpn-7	H-5	*bla* _NDM-1_	307	-	*wzi*173	KL102	-	0
NDM-Kpn-8	H-3	*bla* _NDM-1_	15	14	*wzi*93	KL112	*ybt16*, *ICEKp12*, *iuc1*, *rmp*	4
NDM-Kpn-9	H-3	*bla* _NDM-1_	4853	-	-	KL30	-	0
NDM-Kpn-10	H-3	*bla* _NDM-1_	383	12	-	KL30	*iuc1*, *rmp1*, *KpVP-1*	3
NDM-Kpn-11	H-3	*bla* _NDM-1_	17	-	*wzi*19	KL19	-	0
NDM-Kpn-12	H-8	*bla* _NDM-1_	147	3	*wzi*64	KL64	*ybt9*, *ICEKp3*, *iuc1*, *rmp*	4
NDM-Kpn-13	H-10	*bla* _NDM-1_	383	13	-	KL30	-	0
NDM-Kpn-14	H-5	*bla* _NDM-1_	147	5	*wzi*64	KL64	-	0
NDM-Kpn-15	H-10	*bla* _NDM-1_	383	13	-	KL30	-	0
NDM-Kpn-16	H-5	*bla* _NDM-1_	147	5	*wzi*64	KL64	-	0
NDM-Kpn-17	H-6	*bla* _NDM-1_	29	-	*wzi*115	KL54	*ybt10*, *ICEKp4*	1
NDM-Kpn-18	H-5	*bla* _NDM-1_	383	13	-	KL30	-	0
NDM-Kpn-19	H-2	*bla* _NDM-1_	395	8	*wzi*2	KL2	*ybt16*, *ICEKp12*, *iuc1*, *rmp1*, *KpVP-1*	4
NDM-Kpn-20	H-2	*bla* _NDM-1_	395	8	*wzi*2	KL2	*ybt16*, *ICEKp12*, *iuc1*, *rmp1*, *KpVP-1*	4
NDM-Kpn-21	H-5	*bla* _NDM-1_	147	3	*wzi*64	KL64	*ybt9*, *ICEKp3*, *iuc1*, *rmp1*, *KpVP-1*	4
NDM-Kpn-22	H-6	*bla* _NDM-1_	11	10	*wzi*24	KL24	*ybt15*, *ICEKp11*	1
NDM-Kpn-23	H-4	*bla* _NDM-5_	147	2	*wzi*420	KL10	-	0
NDM-Kpn-24	H-5	*bla* _NDM-1_	395	8	*wzi*2	KL2	*ybt16*, *ICEKp12*, *iuc1*, *rmp1*, *KpVP-1*	4
NDM-Kpn-25	H-4	*bla* _NDM-5_	147	2	*wzi*420	KL10	-	0
NDM-Kpn-26	H-4	*bla* _NDM-5_	147	2	*wzi*420	KL10	-	0
NDM-Kpn-27	H-4	*bla* _NDM-5_	147	2	*wzi*420	KL10	-	0
NDM-Kpn-28	H-1	*bla* _NDM-1_	11	10	*wzi*24	KL24	*ybt15*, *ICEKp11*	1
NDM-Kpn-29	H-4	*bla* _NDM-5_	147	2	*wzi*420	KL10	-	0
NDM-Kpn-30	H-2	*bla* _NDM-1_	395	8	*wzi*2	KL2	*ybt16*, *ICEKp12*, *iuc1*, *rmp1*, *KpVP-1*	4
NDM-Kpn-31	H-1	*bla* _NDM-1_	147	3	*wzi*64	KL64	*ybt9*, *ICEKp3*, *iuc1*, *rmp1*, *KpVP-1*	4
NDM-Kpn-32	H-1	*bla* _NDM-1_	147	3	*wzi*64	KL64	*ybt9*, *ICEKp3*, *iuc1*, *rmp1*, *KpVP-1*	4
NDM-Kpn-33	H-1	*bla* _NDM-1_	512	11	*wzi*154	KL107	-	0
NDM-Kpn-34	H-9	*bla* _NDM-1_	512	11	*wzi*154	KL107	*ybt9*, *ICEKp3*	1
NDM-Kpn-35	H-1	*bla* _NDM-1_	395	7	*wzi*2	KL2	*ybt16*, *ICEKp12*, *iuc1*	4
NDM-Kpn-36	H-13	*bla* _NDM-1_	11	10	*wzi*24	KL24	*ybt15*, *ICEKp11*	1
NDM-Kpn-37	H-1	*bla* _NDM-1_	11	10	*wzi*24	KL24	*ybt15*, *ICEKp11*	1
NDM-Kpn-38	H-9	*bla* _NDM-1_	512	11	*wzi*154	KL107	*ybt9*, *ICEKp3*	1
NDM-Kpn-39	H-2	*bla* _NDM-1_	11	-	*wzi*50	KL15	*ybt9*, *ICEKp3*	1
NDM-Kpn-40	H-6	*bla* _NDM-1_	23	-	*wzi*77	KL57	*ybt9*, *ICEKp3*, *iuc1*, *rmp1*, *KpVP-1*	4
NDM-Kpn-41	H-2	*bla* _NDM-1_	147	3	*wzi*64	KL64	*ybt9*, *ICEKp3*, *iuc1*, *rmp1*, *KpVP-1*	4
NDM-Kpn-42	H-1	*bla* _NDM-1_	11	10	*wzi*24	KL24	*ybt15*, *ICEKp11*	1
NDM-Kpn-43	H-15	*bla* _NDM-1_	11	10	*wzi*24	KL24	*ybt15*, *ICEKp11*	1
NDM-Kpn-44	H-9	*bla* _NDM-1_	512	11	*wzi*154	KL107	-	0
NDM-Kpn-45	H-2	*bla* _NDM-1_	147	4	*wzi*64	KL64	*ybt9*, *ICEKp3*	1
NDM-Kpn-46	H-2	*bla* _NDM-1_	147	4	*wzi*64	KL64	*ybt9*, *ICEKp3*	1
NDM-Kpn-47	H-2	*bla* _NDM-5_	11	9	*wzi*75	KL105	*ybt9*, *ICEKp3*	1
NDM-Kpn-48	H-2	*bla* _NDM-1_	147	4	*wzi*64	KL64	*ybt9*, *ICEKp3*	1
NDM-Kpn-49	H-4	*bla* _NDM-1_	147	3	*wzi*64	KL64	*ybt9*, *ICEKp3*	1
NDM-Kpn-50	H-2	*bla* _NDM-5_	11	9	*wzi*75	KL105	*ybt9*, *ICEKp3*	1
NDM-Kpn-51	H-2	*bla* _NDM-1_	147	4	*wzi*64	KL64	*ybt9*, *ICEKp3*	1
NDM-Kpn-52	H-2	*bla* _NDM-5_	11	9	*wzi*75	KL105	*ybt9*, *ICEKp3*	1
NDM-Kpn-53	H-1	*bla* _NDM-1_	395	7	*wzi*2	KL2	*ybt16*, *ICEKp12*, *iuc1*	4
NDM-Kpn-54	H-1	*bla* _NDM-1_	15	15	*wzi*93	KL112	*ybt16*, *ICEKp12*	1
NDM-Kpn-55	H-3	*bla* _NDM-5_	147	-	*wzi*64	KL64	*ybt9*, *ICEKp3*	1
NDM-Kpn-56	H-1	*bla* _NDM-1_	147	1	*wzi*420	KL10	*ybt10*, *ICEKp4*	1
NDM-Kpn-57	H-1	*bla* _NDM-1_	395	7	*wzi*2	KL2	*ybt16*, *ICEKp12*, *iuc1*	4
NDM-Kpn-58	H-1	*bla* _NDM-1_	15	15	*wzi*93	KL112	*ybt16*, *ICEKp12*	1
NDM-Kpn-59	H-12	*bla* _NDM-1_	147	3	*wzi*64	KL64	*ybt9*, *ICEKp3*, *iuc1*, *rmp1*, *KpVP-1*	4
NDM-Kpn-60	H-11	*bla* _NDM-1_	147	3	*wzi*64	KL64	*ybt9*, *ICEKp3*, *iuc1*, *rmp1*, *KpVP-1*	4
NDM-Kpn-61	H-8	*bla* _NDM-1_	147	3	*wzi*64	KL64	*ybt9*, *ICEKp3*, *iuc1*, *rmp1*, *KpVP-1*	4
NDM-Kpn-62	H-8	*bla* _NDM-1_	11	10	*wzi*24	KL24	*ybt15*, *ICEKp11*	1
NDM-Kpn-63	H-1	*bla* _NDM-1_	11	10	*wzi*24	KL24	*ybt15*, *ICEKp11*	1
NDM-Kpn-64	H-1	*bla* _NDM-1_	1805	6	-	KL48	*iuc1*	3
NDM-Kpn-65	H-1	*bla* _NDM-1_	147	1	*wzi*420	KL10	*ybt10*, *ICEKp4*	1
NDM-Kpn-66	H-7	*bla* _NDM-1_	147	3	*wzi*64	KL64	*ybt9*, *ICEKp3*, *iuc1*, *rmp1*, *KpVP-1*	4
NDM-Kpn-67	H-7	*bla* _NDM-1_	147	3	*wzi*64	KL64	*ybt9*, *ICEKp3*, *iuc1*, *rmp1*, *KpVP-1*	4
NDM-Kpn-68	H-5	*bla* _NDM-1_	147	3	*wzi*64	KL64	*ybt9*, *ICEKp3*, *iuc1*, *rmp1*, *KpVP-1*	4
NDM-Kpn-69	H-7	*bla* _NDM-1_	147	3	*wzi*64	KL64	*ybt9*, *ICEKp3*, *iuc1*, *rmp1*, *KpVP-1*	4
NDM-Kpn-70	H-7	*bla* _NDM-1_	11	10	*wzi*24	KL24	*ybt15*, *ICEKp11*	1
NDM-Kpn-71	H-14	*bla* _NDM-1_	1805	6	-	KL48	*iuc1*	3
NDM-Kpn-72	H-2	*bla* _NDM-5_	147	-	*wzi*420	KL10	-	0
NDM-Kpn-73	H-1	*bla* _NDM-1_	11	10	*wzi*24	KL24	*ybt15*, *ICEKp11*	1
NDM-Kpn-74	H-6	*bla* _NDM-1_	147	1	*wzi*420	KL24	*ybt10*, *ICEKp4*	1
NDM-Kpn-75	H-1	*bla* _NDM-1_	147	3	*wzi*64	KL64	*ybt9*, *ICEKp3*	1
NDM-Kpn-76	H-1	*bla* _NDM-1_	1805	6	-	KL48	*iuc1*	3
NDM-Kpn-77	H-1	*bla* _NDM-1_	147	1	*wzi*420	KL10	*ybt10*, *ICEKp4*	1
NDM-Kpn-78	H-1	*bla* _NDM-5_	234	-	*wzi*272	KL30	-	0
NDM-Kpn-79	H-1	*bla* _NDM-1_	1805	6	-	KL48	*iuc1*	3
NDM-Kpn-80	H-1	*bla* _NDM-1_	1117	-	*wzi356*	KL117	-	0
NDM-Kpn-81	H-17	*bla* _NDM-1_	147	3	*wzi*64	KL64	*ybt9*, *ICEKp3*, *iuc1*, *rmp1*, *KpVP-1*	4
NDM-Kpn-82	H-5	*bla* _NDM-1_	147	1	*wzi*420	KL10	*ybt1*, *ICEKp4*	1
NDM-Kpn-83	H-6	*bla* _NDM-1_	147	1	*wzi*420	KL10	*ybt1*, *ICEKp4*	1
NDM-Kpn-84	H-2	*bla* _NDM-1_	11	10	*wzi*24	KL24	*ybt15*, *ICEKp11*	1
NDM-Kpn-85	H-6	*bla* _NDM-1_	147	1	*wzi*420	KL10	*ybt1*, *ICEKp4*	1
NDM-Kpn-86	H-4	*bla* _NDM-1_	147	3	*wzi*64	KL64	*ybt9*, *ICEKp3*, *iuc1*, *rmp1*, *KpVP-1*	4
NDM-Kpn-87	H-2	*bla* _NDM-1_	147	3	*wzi*64	KL64	*ybt9*, *ICEKp3*	1
NDM-Kpn-88	H-2	*bla* _NDM-1_	147	3	*wzi*64	KL64	*ybt9*, *ICEKp3*	1
NDM-Kpn-89	H-1	*bla* _NDM-1_	11	10	*wzi*24	KL24	*ybt15*, *ICEKp11*	1
NDM-Kpn-90	H-1	*bla* _NDM-1_	147	3	*wzi*64	KL64	*ybt9*, *ICEKp3*	1
NDM-Kpn-91	H-16	*bla* _NDM-1_	147	1	*wzi*420	KL10	*ybt1*, *ICEKp4*	1
NDM-Kpn-92	H-17	*bla* _NDM-1_	1805	6	-	KL48	*iuc1*	3
NDM-Kpn-93	H-1	*bla* _NDM-1_	147	3	*wzi*64	KL64	*ybt9*, *ICEKp3*	1
NDM-Kpn-94	H-3	*bla* _NDM-1_	147	3	*wzi*64	KL64	*ybt9*, *ICEKp3*, *iuc1*, *rmp1*, *KpVP-1*	4
NDM-Kpn-95	H-17	*bla* _NDM-1_	1805	6	-	KL48	*iuc1*	3
NDM-Kpn-96	H-16	*bla* _NDM-1_	147	1	*wzi*420	KL10	*ybt1*, *ICEKp4*	1
NDM-Kpn-97	H-19	*bla* _NDM-1_	3299	-	*wzi*2	KL2	*ybt16*, *ICEKp12*	1
NDM-Kpn-98	H-2	*bla* _NDM-1_	11	10	*wzi*24	KL24	*ybt15*, *ICEKp11*	1
NDM-Kpn-99	H-1	*bla* _NDM-1_	11	10	*wzi*24	KL24	*ybt15*, *ICEKp11*	1
NDM-Kpn-100	H-2	*bla* _NDM-5_	4081	-	-	KL125	-	0
NDM-Kpn-101	H-2	*bla* _NDM-1_	395	-	*wzi*160	KL39	*Ybt16*, *ICEKp12*	1
NDM-Kpn-102	H-2	*bla* _NDM-1_	147	3	*wzi*64	KL64	*ybt9*, *ICEKp3*	1
NDM-Kpn-103	H-1	*bla* _NDM-1_	147	1	*wzi*420	KL10	*ybt1*, *ICEKp4*	1
NDM-Kpn-104	H-7	*bla* _NDM-1_	11	10	*wzi*24	KL24	*ybt15*, *ICEKp11*	1
NDM-Kpn-105	H-7	*bla* _NDM-1_	1805	-	-	KL48	-	0
NDM-Kpn-106	H-7	*bla* _NDM-1_	11	10	*wzi*24	KL24	*ybt15*, *ICEKp11*	1
NDM-Kpn-107	H-7	*bla* _NDM-5_	383	-	-	KL30	*iuc1*	3
NDM-Kpn-108	H-7	*bla* _NDM-1_	11	10	*wzi*24	KL24	*ybt15*, *ICEKp11*	1
NDM-Kpn-109	H-7	*bla* _NDM-1_	11	10	*wzi*24	KL24	*ybt15*, *ICEKp11*	1
NDM-Kpn-110	H-7	*bla* _NDM-1_	11	10	*wzi*24	KL24	*ybt15*, *ICEKp11*	1
NDM-Kpn-111	H-7	*bla* _NDM-5_	147	3	*wzi*64	KL64	*ybt9*, *ICEKp3*	1
NDM-Kpn-112	H-7	*bla* _NDM-5_	147	3	*wzi*64	KL64	*ybt9*, *ICEKp3*, *iuc1*, *rmp1*, *KpVP-1*	4
NDM-Kpn-113	H-7	*bla* _NDM-1_	147	1	*wzi*420	KL10	*ybt1*, *ICEKp4*	1
NDM-Kpn-114	H-7	*bla* _NDM-1_	11	10	*wzi*24	KL24	*ybt15*, *ICEKp11*	1
NDM-Kpn-115	H-7	*bla* _NDM-1_	11	10	*wzi*24	KL24	*ybt15*, *ICEKp11*	1
NDM-Kpn-116	H-7	*bla* _NDM-1_	147	3	*wzi*64	KL64	*ybt9*, *ICEKp3*	1
NDM-Kpn-117	H-4	*bla* _NDM-5_	14	-	*wzi*2	KL2	-	0
NDM-Kpn-118	H-1	*bla* _NDM-1_	11	10	*wzi*24	KL24	*ybt15*, *ICEKp11*	1
NDM-Kpn-119	H-18	*bla* _NDM-1_	147	3	*wzi*64	KL64	*ybt9*, *ICEKp3*, *iuc1*, *rmp1*	4
NDM-Kpn-120	H-1	*bla* _NDM-1_	147	3	*wzi*64	KL64	*ybt9*, *ICEKp3*, *iuc1*, *rmp1*, *KpVP-1*	4
NDM-Kpn-121	H-4	*bla* _NDM-1_	147	1	*wzi*420	KL10	*ybt1*, *ICEKp4*	1
NDM-Kpn-122	H-6	*bla* _NDM-1_	395	-	*wzi*2	KL2	*ybt16*, *ICEKp12*, *iuc1*	4
NDM-Kpn-123	H-2	*bla* _NDM-1_	147	3	*wzi*64	KL64	*ybt9*, *ICEKp3*	1
NDM-Kpn-124	H-7	*bla* _NDM-5_	6118	-	-	KL30	*iuc1*	3
NDM-Kpn-125	H-7	*bla* _NDM-1_	11	10	*wzi*24	KL24	*ybt15*, *ICEKp11*	1
NDM-Kpn-126	H-7	*bla* _NDM-1_	147	1	*wzi*420	KL10	*ybt1*, *ICEKp4*	1

^1^ Strains collected from 19 different hospitals in Rome (H-1 to H-19); ^2^ Sequence Type (ST) identified by MLST; ^3^ Cluster Types (CT-1 to CT-15) identified using cgMLST method; ^4^ Capsule Type (KL); ^5^ Virulence score range from 0 to 5. The virulence scores were defined based on the presence or absence of these loci, as follows: 0  =  no yersinabactin, colibactin or aerobactin; 1  =  yersiniabactin only; 2  =  yersiniabactin and colibactin (or colibactin only); 3  =  aerobactin without yersiniabactin or colibactin; 4  =  aerobactin with yersiniabactin (no colibactin); and 5  =  yersiniabactin, colibactin, and aerobactin.

**Table 2 antibiotics-13-01045-t002:** Distribution and virulence scores of Kpn-NDM isolates grouped in the CTs.

Cluster Types (CTs)	Hospitals (H)	Virulence Score				Sequence Types (STs)
		0	1	3	4	
1	H-1, H-4, H-5, H-6, H-7, H16	-	13	-	-	147
2	H-4	5	-	-	-	147
3	H1, H-2, H-3, H-4, H-5, H-7, H-8, H-11, H-12, H-17, H-18	-	10	-	19	147
4	H-2	-	4	-	-	147
5	H-5	2	-	-	-	147
6	H-1, H-14, H-17	-	-	6	-	1805
7	H-1	-	-	-	3	395
8	H-2, H-5	-	-	-	4	395
9	H-2	-	3	-	-	11
10	H-1, H-2, H-4, H-6, H-7, H-8, H-13, H-15	-	25	-	-	11
11	H-1, H-9	2	2	-	-	512
12	H-3	-	-	2	-	383
13	H-5, H-10	3	-	-	-	383
14	H-3	-	-	-	3	15
15	H-1	-	2	-	-	15
-	H-1, H-2, H-3, H-4, H-5, H-6, H-7, H-19	9	5	2	2	11, 14, 17, 23, 29, 147, 234, 307, 383, 395, 1117, 1805, 3299, 4081, 4853, 6118
Total per Virulence Score		21	64	10	31	

## Data Availability

The original data presented in this study can be found in the Excel database created ad hoc, which is archived at the authors’ institution (INMI L. Spallanzani I.R.C.C.S., Rome, Italy). The whole-genome sequencing files were submitted to NCBI BioProjects ID PRJNA1078116 and ID PRJNA1125835.

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
