# Peer review of "Molecular Characterization of Multidrug-Resistant and Hypervirulent New Delhi Metallo-Beta-Lactamase Klebsiella pneumoniae in Lazio, Italy: A Five-Year Retrospective Study"

_antibiotics, 2024, doi:10.3390/antibiotics13111045_

Round 1
Reviewer 1 Report
Comments and Suggestions for Authors
The manuscript titled "Molecular Characterization of Multidrug-Resistant and Hyper-virulent NDM-Klebsiella pneumoniae in Lazio, Italy: A Five-Year Retrospective Study," is well written and is relevant to the scientific community as well as public health authorities.
The manuscript has the following strengths:
1. Comprehensive data collection spanning five years from a significant number of hospitals adds a lot of weight to the findings.
2. The use of WGS and molecular typing is appropriate and provides valuable insights into the genetic makeup of the bacteria and its resistance and virulence profiles.
3. The study emphasizes antimicrobial resistance which is crucial, especially given the rising threat of resistant strains in healthcare settings.
The manuscript can however benefit from addressing the following major concerns:
1. Results section - Clarity of results
The results lack clarity in their presentation, for instance the manuscript frequently mentions supplementary data which makes it difficult for readers to follow, thus it is suggested that important information or data presented in the supplementary data be incorporated in the main text such as the minimum spanning network must be presented in the main text, of course after improving its resolution. Further, I propose that some of the supplementary tables be converted to user friendly visual images for easier understanding by the readers, some R packages can be used for this.
I strongly recommend that the authors build a phylogenetic tree using whole-genome sequencing data or multilocus sequence typing (MLST) data already gathered and then annotate the tree with the following information:
1. Resistance profiles (e.g., presence of NDM-1, NDM-5, ESBL genes).
2. Virulence factors (e.g., rmpA, iuc, yersiniabactin genes).
3. Virulence scores for each strain.
4. Geographical origin (hospital or town or district if possible).
5. Temporal data (year of isolation).
And then color-code specific clusters (CTs and STs) that correspond to high-risk clones and emphasize any groups with significant resistance and virulence characteristics. This will greatly enhance the appeal and presentation of data in this manuscript.
2. Discussion section - Lack of detailed clinical impact
While the study highlights the prevalence of multidrug resistance and hypervirulence, it does not fully explain how these findings might impact clinical treatment or patient outcomes. For example, what were the clinical outcomes for patients infected with strains showing the highest virulence scores? How did the different resistance profiles influence treatment choices? A more in-depth discussion of the clinical implications of the study findings is thus suggested and where possible, kindly provide a breakdown of patient outcomes and relate these findings to the therapeutic challenges faced by health care providers. This will make the research more directly relevant to clinical practice.
Although the study is limited to Lazio in Italy, the authors should consider discussing the global applicability of the findings in greater detail. For example, the manuscript briefly mentions the spread of NDM globally, but this could be expanded a bit more to show how Lazio compares to other regions in Italy or globally, basically highlighting how this study’s results fit into the broader global picture of antimicrobial resistance with special interest of course to NDM-K. pneumoniae outbreaks.
The findings from the virulence scores are critical, however the implications of the scores are not conclusively explained such as what makes them significant, and how do they influence the potential for epidemic spread or patient mortality? Suggest adding a discussion on the clinical and epidemiological implications of the virulence profiles in this study and providing comparisons with other hypervirulent strains documented in literature elsewhere.
The conclusion would benefit from more specific recommendations for example, what practical steps must be taken to prevent the spread of high-risk clones in Italy and how can this study inform policy changes to infection control protocols in hospitals? The authors can consider proposing specific actionable steps for healthcare providers and policymakers, such as screening strategies, hospital-level infection control measures, or international protocols for monitoring and reporting cases.
The limitations section mentions the regional focus of the study and potential underreporting. While this is fine, it would also be helpful to discuss the impact of these limitations on the results. If some labs lacked the means to characterize isolates, how could this have affected the study's conclusions with regards to the spread of resistant strains? Also suggest potential avenues for future research such as expanding surveillance to other regions or improving laboratory capacity to address the any gaps.
Incorporating these improvements will make the manuscript more impactful and valuable for both health care and researchers.
Comments on the Quality of English Language
The quality of the english language is adequate.
Author Response
Reviewer 1
Point-by-point response
Open Review
Comments and Suggestions for Authors
The manuscript titled "Molecular Characterization of Multidrug-Resistant and Hyper-virulent NDM-Klebsiella pneumoniae in Lazio, Italy: A Five-Year Retrospective Study," is well written and is relevant to the scientific community as well as public health authorities.
The manuscript has the following strengths:
- Comprehensive data collection spanning five years from a significant number of hospitals adds a lot of weight to the findings.
- The use of WGS and molecular typing is appropriate and provides valuable insights into the genetic makeup of the bacteria and its resistance and virulence profiles.
- The study emphasizes antimicrobial resistance which is crucial, especially given the rising threat of resistant strains in healthcare settings.
The manuscript can however benefit from addressing the following major concerns:
- Results section - Clarity of results
The results lack clarity in their presentation, for instance the manuscript frequently mentions supplementary data which makes it difficult for readers to follow, thus it is suggested that important information or data presented in the supplementary data be incorporated in the main text such as the minimum spanning network must be presented in the main text, of course after improving its resolution. Further, I propose that some of the supplementary tables be converted to user friendly visual images for easier understanding by the readers, some R packages can be used for this.
I strongly recommend that the authors build a phylogenetic tree using whole-genome sequencing data or multilocus sequence typing (MLST) data already gathered and then annotate the tree with the following information:
- Resistance profiles (e.g., presence of NDM-1, NDM-5, ESBL genes).
- Virulence factors (e.g., rmpA, iuc, yersiniabactingenes).
- Virulence scores for each strain.
- Geographical origin (hospital or town or district if possible).
- Temporal data (year of isolation).
And then color-code specific clusters (CTs and STs) that correspond to high-risk clones and emphasize any groups with significant resistance and virulence characteristics. This will greatly enhance the appeal and presentation of data in this manuscript.
Reply:
Following both of the reviewers’ comments, and in an effort to keep the article as readable as possible, we moved Figure S1 from supplementary data to the main text (now Figure 1), and we added the information on STs to Table S3 which was moved to the main text and is now Table 1. We also added a supplementary Table (S3) to put together phenotypic profiles, resistance profiles and STs.
The suggestion to convert the tables to visual images is very interesting, however we found it difficult with so many data and variables.
We also followed the suggestion of building a phylogenetic tree, however the high amount of data did not allow us to produce a figure with an acceptable level of resolution. We therefore included only the most significant CTs and thought it could be provided as a supplementary figure (Figure S1).
- Discussion section - Lack of detailed clinical impact
While the study highlights the prevalence of multidrug resistance and hypervirulence, it does not fully explain how these findings might impact clinical treatment or patient outcomes. For example, what were the clinical outcomes for patients infected with strains showing the highest virulence scores? How did the different resistance profiles influence treatment choices? A more in-depth discussion of the clinical implications of the study findings is thus suggested and where possible, kindly provide a breakdown of patient outcomes and relate these findings to the therapeutic challenges faced by health care providers. This will make the research more directly relevant to clinical practice.
Reply:
This is a very important point. However, another limitation of our study is the lack of clinical data on the patients. Being a laboratory surveillance, we have no information on the type of treatment carried out, the effectiveness of the treatment and the outcome of each patient. We mentioned this as another limitation of our study (Page 11 lines 250-252). We added references (17, 18, 19, 20, 21) that address the clinical relevance of hvKp in the introduction and modified the text accordingly (Page 2 lines 78-81).
Although the study is limited to Lazio in Italy, the authors should consider discussing the global applicability of the findings in greater detail. For example, the manuscript briefly mentions the spread of NDM globally, but this could be expanded a bit more to show how Lazio compares to other regions in Italy or globally, basically highlighting how this study’s results fit into the broader global picture of antimicrobial resistance with special interest of course to NDM-K. pneumoniae outbreaks.
Reply:
To answer this comment, we added the following references in the Introduction: 7, 8
The reviewer is correct in pointing out the weakness of the regional nature of our study, which we have already mentioned as a limitation of our study (Page 11 lines 242-243). It is not possible however to expand the study to other regions because of the regional organization of the Italian National Health Service, where each region is autonomous in their surveillance activities. Our study was conducted following the outbreak in Tuscany between 2018 and 2019 (Page 2 lines 62-64, Ref. 9), and our results show that the main ST found (ST147) is the same one found in the Tuscan outbreak. The blaNDM-1 variant also appears to be the most represented in both studies. From these results it can be deduced that the cross-section that emerged in the Lazio region coincides with that found in the Tuscany region. We have rephrased the limitations paragraph to further clarify this point (Page 11 lines 242-243).
The findings from the virulence scores are critical, however the implications of the scores are not conclusively explained such as what makes them significant, and how do they influence the potential for epidemic spread or patient mortality? Suggest adding a discussion on the clinical and epidemiological implications of the virulence profiles in this study and providing comparisons with other hypervirulent strains documented in literature elsewhere.
Reply:
We modified the paragraph to add some information on the importance of the virulence genes found, ad included Ref. 17, 18, 19, 20, 21.
The conclusion would benefit from more specific recommendations for example, what practical steps must be taken to prevent the spread of high-risk clones in Italy and how can this study inform policy changes to infection control protocols in hospitals? The authors can consider proposing specific actionable steps for healthcare providers and policymakers, such as screening strategies, hospital-level infection control measures, or international protocols for monitoring and reporting cases.
Reply:
This is a very important point. We believe the key is a timely and effective surveillance in hospitals and the community in a one-health approach, employing the newest technology. The critical aspects are international collaborations, resources and funding for surveillance and infection control, and further research into new therapeutic options (Page 12 Lines 312-314).
The limitations section mentions the regional focus of the study and potential underreporting. While this is fine, it would also be helpful to discuss the impact of these limitations on the results. If some labs lacked the means to characterize isolates, how could this have affected the study's conclusions with regards to the spread of resistant strains? Also suggest potential avenues for future research such as expanding surveillance to other regions or improving laboratory capacity to address the any gaps.
Incorporating these improvements will make the manuscript more impactful and valuable for both health care and researchers.
Reply:
The fact that some laboratories do not have the necessary means to be able to characterise bacterial isolates means that these bacterial isolates are not sent to our laboratory and are therefore escape surveillance. This results in a possible underestimation of NDM isolates (Page 11 lines 244-247). In a broader view, an underestimation of MDR bacterial isolates may result in a higher prevalence than estimated.
The Italian healthcare system is regional-based, so each region has its own reference centre to which bacterial isolates are sent, but certainly a better coordination between regions and at the international level is the way forward (Page 12 lines 312-314).
Reviewer 2 Report
Comments and Suggestions for Authors
The manuscript titled "Molecular Characterization of Multidrug-Resistant and Hypervirulent NDM-Klebsiella pneumoniae in Lazio, Italy: A Five-Year Retrospective Study" appears to be a well-conducted molecular epidemiology study that provides data on the concerning spread of highly resistant and virulent NDM-Kpn strains in this region. The findings highlight the importance of ongoing surveillance and the need for coordinated efforts to control these pathogens.
Comments for more results and/or discussions:
- The study is conducted in the Lazio region of Italy, which may limit its geographic scope. If the authors consider expanding to compare with other regions in Italy or in Europe regarding NDM-Kpn spreading, it can strengthen the discussion.
- The study shows a 5-year collection of NDM-Kpn data, but there's a lack of results and/or discussion on the changing trend of NDM-Kpn during this period.
- The authors should add more comparative analysis from the dataset (e.g., ST vs. resistant phenotypes, resistance genes vs. phenotypic resistance, or merge to analyze together). This can present an overall of the data to readers and enhance the strong presentation of this study.
- In the introduction or results section, the authors can provide more background on virulence genes to improve understanding and make it easier for readers to follow, including highlighting the main findings and linking them to phenotypes.
- One aspect to clarify in the methods section is the identification of NDM-producing strains. The authors should specify the cut-off value for ceftazidime/avibactam used to classify resistant strains and provide the reference (EUCAST). A brief description of the molecular diagnostic methods used for confirmation should also be included in this part.
- Figure S1 can be moved to the main text to show an overview of sequence types in the study.
- Table 1 should add one column to show which cgMLSTs are contained in each CT to provide more detailed epidemiological data.
Author Response
Reviewer 2
Point-by-point response
Comments and Suggestions for Authors
The manuscript titled "Molecular Characterization of Multidrug-Resistant and Hypervirulent NDM-Klebsiella pneumoniae in Lazio, Italy: A Five-Year Retrospective Study" appears to be a well-conducted molecular epidemiology study that provides data on the concerning spread of highly resistant and virulent NDM-Kpn strains in this region. The findings highlight the importance of ongoing surveillance and the need for coordinated efforts to control these pathogens.
Comments for more results and/or discussions:
- The study is conducted in the Lazio region of Italy, which may limit its geographic scope. If the authors consider expanding to compare with other regions in Italy or in Europe regarding NDM-Kpn spreading, it can strengthen the discussion.
Reply:
The reviewer is correct in pointing out this weakness, which we have already mentioned as a limitation of our study (Page 11 lines 239-243). It is not possible however to expand the study to other regions because of the regional organization of the Italian National Health Service, where each region is autonomous in their surveillance activities. Our study was conducted following the outbreak in Tuscany between 2018 and 2019 (Page 2 lines 62-64, Ref. 9), and our results show that the main ST found (ST147) is the same one found in the Tuscan outbreak. The blaNDM-1 variant also appears to be the most represented in both studies. From these results it can be deduced that the cross-section that emerged in the Lazio region coincides with that found in the Tuscany region. We have rephrased the limitations paragraph to further clarify this point (Page 11 lines 239-243)
- The study shows a 5-year collection of NDM-Kpn data, but there's a lack of results and/or discussion on the changing trend of NDM-Kpn during this period.
Reply:
It would be very interesting to include data on changing patterns over time. We analysed our data with that objective in mind, but we did not see any evicence of a changing trend. Moreover, our data are biased by higher numbers of isolates received in more recent years (e.g. 11 in 2019, 7 in 2020 and 24 and 71 in 2022 and 2023), likely reflecting the improvement in the laboratories’ efficiency in collecting and sending the strains, after the difficult years of the Covid-19 pandemic.
We added this factor in the limitations paragraph (Page 11 lines 246-249)
- The authors should add more comparative analysis from the dataset (e.g., ST vs. resistant phenotypes, resistance genes vs. phenotypic resistance, or merge to analyze together). This can present an overall of the data to readers and enhance the strong presentation of this study.
Reply:
To address the reviewers’s comment, we designed a new Table (Supplementary Table S3), putting together data of resistance (phenotypic or presence of genes) and ST. Analysing the data, we found that a specific resistance genotype may not always align with the phenotype. This means that the presence of a resistance mutation or gene does not guarantee the development of phenotypic resistance. This disconnect between genotype and phenotype has significant implications for understanding the mechanisms of resistance and developing effective treatment strategies, as reported also by the newly added Ref. 36.
We included this information in the text (Page 11 Lines 224-226).
- In the introduction or results section, the authors can provide more background on virulence genes to improve understanding and make it easier for readers to follow, including highlighting the main findings and linking them to phenotypes.
Reply:
Following the reviewer’s suggestion, we modified the text of the Introduction (Page 2 Lines 78-81) and added the following references: 17, 18, 19, 20, 21.
- Liu, Y. C., Cheng, D. L., & Lin, C. L. (1986). Klebsiella pneumoniae liver abscess associated with septic endophthalmitis. Archives of internal medicine, 146(10), 1913–1916.
- Siu, L. K., Fung, C. P., Chang, F. Y., Lee, N., Yeh, K. M., Koh, T. H., & Ip, M. (2011). Molecular typing and virulence analysis of serotype K1 Klebsiella pneumoniae strains isolated from liver abscess patients and stool samples from noninfectious subjects in Hong Kong, Singapore, and Taiwan. Journal of clinical microbiology, 49(11), 3761–3765. https://doi.org/10.1128/JCM.00977-11.
- Kochan, T. J., Nozick, S. H., Medernach, R. L., Cheung, B. H., Gatesy, S. W. M., Lebrun-Corbin, M., Mitra, S. D., Khalatyan, N., Krapp, F., Qi, C., Ozer, E. A., & Hauser, A. R. (2022). Genomic surveillance for multidrug-resistant or hypervirulent Klebsiella pneumoniae among United States bloodstream isolates. BMC infectious diseases, 22(1), 603. https://doi.org/10.1186/s12879-022-07558-1.
- Zhang, Y., Zeng, J., Liu, W., Zhao, F., Hu, Z., Zhao, C., Wang, Q., Wang, X., Chen, H., Li, H., Zhang, F., Li, S., Cao, B., & Wang, H. (2015). Emergence of a hypervirulent carbapenem-resistant Klebsiella pneumoniae isolate from clinical infections in China. The Journal of infection, 71(5), 553–560. https://doi.org/10.1016/j.jinf.2015.07.010.
- Su, S. C., Siu, L. K., Ma, L., Yeh, K. M., Fung, C. P., Lin, J. C., & Chang, F. Y. (2008). Community-acquired liver abscess caused by serotype K1 Klebsiella pneumoniae with CTX-M-15-type extended-spectrum beta-lactamase. Antimicrobial agents and chemotherapy, 52(2), 804–805. https://doi.org/10.1128/AAC.01269-07.
- One aspect to clarify in the methods section is the identification of NDM-producing strains. The authors should specify the cut-off value for ceftazidime/avibactam used to classify resistant strains and provide the reference (EUCAST). A brief description of the molecular diagnostic methods used for confirmation should also be included in this part.
Reply:
According to EUCAST guidelines, MIC breakpoints for Enterobacterales for ceftazidime-avibactam is 8 mg/L (S ≤ and R >). Our study was conducted starting from strains that were found to be resistant to CZA, collected in the framework of a regional surveillance scheme. Given that resistance to CZA could be due to the presence of a mutated KPC or a metallo beta-lactamase, as an initial screening we performed an immunochromatographic test, and only NDM isolates were included in the study. The results were confirmed by WGS. We have clarified the selection procedure in the methods section (Page 12 lines 282-284).
- Figure S1 can be moved to the main text to show an overview of sequence types in the study.
Reply:
Figure S1 was moved to the main text and is now Figure 1
- Table 1 should add one column to show which cgMLSTs are contained in each CT to provide more detailed epidemiological data.
Reply:
Done. The Table is now named Table 2
Round 2
Reviewer 2 Report
Comments and Suggestions for Authors
This version is accepted. No comments.